# WF-Bench: A Benchmark for Neural Network WaveFunction Expressivity and Scaling Laws

Lixing Zhang [1]   Guijing Duan [2]   Di Luo [3][4]

## Abstract

We present a comprehensive benchmarking dataset and empirical scaling law analysis for neural network wavefunctions by matching them to a wide spectrum of famous many body target wavefunctions. The dataset, WF-Bench, spans multiple distinct regimes of strongly correlated quantum matter, including topological states, Wigner crystals, and superconducting wavefunctions, providing a diverse and challenging test bed for neural network wavefunction expressivity. We introduce a systematic and reproducible benchmarking protocol for target wavefunction matching, enabling consistent performance evaluation across different neural network wavefunction architectures. By using wavefunction fidelity as the uniform metric, we discover empirical scaling laws that characterize how representability depends on system size and key model parameters, including number of determinant and model depth. By applying our benchmark protocol on Psiformer and Ferminet, we show that WF-Bench establishes a unified dataset driven framework for evaluating and comparing neural network wavefunctions and for guiding the design of future architectures.

## 1. Introduction

Recent years have seen rapid progress in neural network (NN) wavefunctions as flexible variational ansätze for quantum many body problems. Harnessing the representation power of deep neural networks, NN wavefunctions have demonstrated remarkable scalability and accuracy across a variety of quantum many body tasks, including ground-state optimization(Ren et al., 2023; Wang et al., 2024), excited-state spectroscopy(Kiyohara et al., 2020; Pfau et al., 2024), open system dynamics(Irikura & Saito, 2020; Luo et al., 2022a), real-time evolutions(Lin et al., 2024; Gutiérrez & Mendl, 2022), and quantum state tomography(Torlai et al., 2018; Quek et al., 2021). These capabilities have broad impact across multiple disciplines, including condensed matter physics(Luo et al., 2024; Levine et al., 2019; Zhang & Luo, 2025), high energy physics(Luo et al., 2022b; 2021), quantum chemistry(Hermann et al., 2023; Barrett et al., 2022), quantum information science(Levine et al., 2017; Shen et al., 2020), and quantum computations(Min-Gang et al., 2023; Benedetti et al., 2019). As the field expands, an increasing number of powerful neural network architectures have been proposed to tackle these challenges. Notable examples include Ferminet (Pfau et al., 2020), based on streaming permutation-equivariant one- and two body features; Psiformer (von Glehn et al., 2022), which leverages self-attention mechanisms; graph neural networks built on message passing (Luo et al., 2025; Pescia et al., 2024; Kim et al., 2024); and DeepSolid (Li et al., 2022), designed for periodic systems.

It has long been known that the exact solution of the many body Schrödinger equation requires computational resources that grow exponentially with the system size. However, approximate solutions can often be obtained with only polynomial computational cost. Therefore, a good NN wavefunction must have a careful balance between computational efficiency and representation accuracy. This is directly related to the expressive power of the underlying architecture. Despite rapid advances in NN wavefunctions, there is still no systematic understanding of how representation power varies across architectures and classes of physical systems, nor a unified benchmarking framework for evaluating NN wavefunction expressivity and scaling laws.

To address this issue, we introduce WF-Bench, a comprehensive dataset for benchmarking the representation power of NN wavefunctions. WF-Bench contains over 30 target wavefunctions spanning three major classes: topological states, superconducting states, and Wigner crystals. They

[1]Department of Chemistry and Biochemistry, University of California Los Angeles, Los Angeles, CA 90095, USA [2]School of Physics and Beijing Key Laboratory of Opto-electronic Functional Materials & Micro-nano Devices, Renmin University of China, Beijing 100872, China [3]Department of Physics, Tsinghua University, Beijing 100084, China [4]Institute of Advanced Study, Tsinghua University, Beijing 100084, China. Correspondence to: Di Luo <diluo@tsinghua.edu.cn>.

*Proceedings of the 43rd International Conference on Machine Learning*, Seoul, South Korea. PMLR 306, 2026. Copyright 2026 by the author(s).

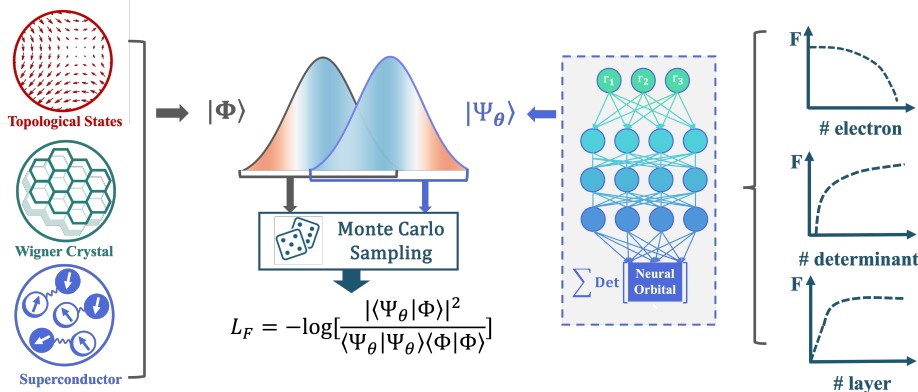

*Figure 1.* Schematic of the WF-Bench workflow. The dataset consists three categories of wavefunctions: topological states, Wigner crystals and superconducting wavefunctions. By using $-\log F$ as the loss function, NN wavefunctions are optimized to match both the phase and the amplitude of the target wavefunctions. This enables uniform representation power benchmarks across key system and model parameters, including number of electron, determinant number, and network depth.

cover a broad range of physically important systems with intrinsically different correlation structures. The topological class includes well-known trial wavefunctions with nontrivial topological order and complex phase structure, such as the Laughlin (Laughlin, 1983) and Moore Read (Moore & Read, 1991) states of fractional quantum Hall systems. The superconducting class consists of Bardeen Cooper Schrieffer (BCS) type wavefunctions with diverse pairing symmetries and spin configurations. The Wigner crystal class comprises states exhibiting spontaneous translational symmetry breaking driven by strong interactions or external periodic potentials. Depending on the target state, fermionic antisymmetry is realized either through a Slater determinant form or a Pfaffian form.

In addition, by optimizing the fidelity between NN wavefunction and the target wavefunctions, we present a uniform benchmarking protocol under the framework of Monte-Carlo (MC) sampling. By applying our benchmarking protocol to two of the widely used NN wavefunction architectures, Psiformer and Ferminet, we systematically study the empirical scaling of the maximum achievable fidelity $F$ with respect to three important parameters: the number of electrons $N_e$, the number of determinant $N_{\text{det}}$, and the number of layers $N_{\text{layer}}$. We found empirically that $1 - F$ follows a power-law scaling with $N_e$ for all wavefunctions in our dataset. As $N_{\text{det}}$ and $N_{\text{layer}}$ increase, the fidelity $F$ exhibits clear diminishing returns, with a sharp initial improvement followed by saturation at larger values.

Contributions of this work are summarized as followed:

- **A comprehensive dataset of physically important systems.** We construct a comprehensive dataset spanning three major classes of quantum many body states—topological states, superconducting wavefunctions, and Wigner crystals—covering a wide range of

correlation strength, pairing symmetry, and phase complexity.

- **A unified benchmarking protocol for NN wavefunction representability.** We propose a unified training and evaluation protocol based on fidelity optimization, enabling fair and reproducible comparisons across different NN wavefunction architectures and expressiveness parameters, such as number of determinants and number of layers.

- **Empirical scaling laws for NN wavefunction representability.** By systematically varying electron number and network capacity, we identify empirical scaling laws governing the achievable fidelity of NN wavefunction, and relate the scaling exponents to physical properties of the target wavefunctions.

## 2. Related Work

**NN wavefunction architecture** In the past few years, there has been a rapid growth of NN wavefunctions designed for different tasks. This includes early works for generic many body ground-state problems (Carleo & Troyer, 2017), as well as autoregressive formulations that enable efficient sampling and scalable optimization (Sharir et al., 2020; Hibat-Allah et al., 2020). Subsequent work has extended NN wavefunctions to finite-temperature settings (Irikura & Saito, 2020), real-time dynamics (Lee et al., 2021), and geometrically motivated representations (Han & Hartnoll, 2020). To incorporate fermionic statistics, several works introduce backflow transformations into NN wavefunctions(Luo & Clark, 2019; Li et al., 2024). More expressive architectures have been proposed using fermionic hidden states (Robledo Moreno et al., 2022), tensor-network based autoregressive networks (Chen et al., 2023), and message-

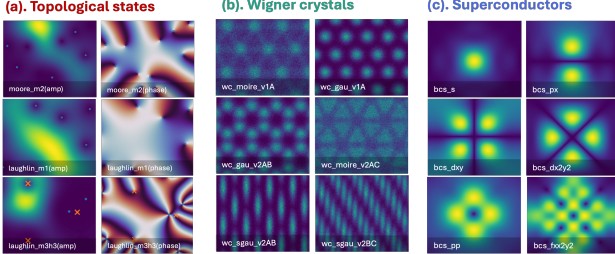

*Figure 2.* Feature plots of target wavefunctions. (a) Amplitude and phase of topological states obtained by scanning the position of one electron while fixing all others. Blue dots indicate fixed electrons, and red crosses denote quasiholes. (b) Charge density patterns of different Wigner crystals. (c) Real space structure of the pairing functions for different superconducting states.

passing networks for homogeneous fermionic gas (Pescia et al., 2024; Kim et al., 2024). Recent works further explore Pfaffian and pairing-based neural wavefunctions (Gao & Günnemann, 2024; Lou et al., 2024), applications to dense and strongly correlated quantum matter (Xie et al., 2023), and large-scale transformer-style architectures for physical systems (Zhdanov et al., 2025).

**NN wavefunction expressivity** There are ongoing efforts, both theoretical and numerical, to characterize the expressivity and scaling behavior of NN wavefunctions across different quantum settings. Theoretical analyses of the representational power and entanglement structure of neural network wavefunctions include (Deng et al., 2017), which shows that neural quantum states can efficiently represent highly entangled many body states, and (Carleo et al., 2018), which demonstrates exact representations of quantum many body wavefunctions using deep neural networks. In (Yang et al., 2024), the conditions under which classical neural networks can represent quantum states are analyzed, while (Paul, 2025) derives rigorous bounds on the entanglement supported by neural network wavefunctions. Numerically, (Jiang et al., 2025) verified that the minimal energy error of NN wavefunctions for quantum chemistry scales as $\mathcal{O}(N_{\mathrm{p}}^{-0.52})$, outperforming traditional methods such as CCSD., while (Nazaryan et al., 2025) studies scaling of wavefunction fidelity in fractional quantum Hall Hamiltonians.Finally, (Lu et al., 2026) established an information-theoretic scaling law with numerical verifications for generic auto-regressive neural quantum states.

Despite a number of studies focused on specific topics of models or regimes, there is still a lack of a uniform, consistent, and comprehensive study of how the representation power of NN wavefunctions scales across different physical systems and architectures. The WF-Bench dataset, together with the accompanying benchmarking protocol, is designed to fill this gap and to advance the field toward a systematic and reproducible framework for studying expressivity

scaling in neural network wavefunction.

---

**Algorithm 1** Benchmarking Protocol
___

**Input:** Electron numbers $\{N_e\}$, target wavefunction $|\Phi\rangle$, initial network parameters $\theta$
**Output:** Fidelity series $\{F(N_e)\}$
Initialize $\theta_{\mathrm{last}} = \theta$
**for** each $N_e \in \{N_e\}$ **do**
   Burn in MC walkers for $100N_e$ steps
   Initialize parameters $\theta \leftarrow \theta_{\mathrm{last}}$
   **if** $|\Phi\rangle$ is a topological state **then**
     **for** $t = 1$ to 20,000 **do**
       Burn in walkers for $5N_e$ steps
       Sample configurations $\{\mathbf{R}\} \sim p_{\mathrm{mix}}$
       Estimate pretraining loss $L_{\mathrm{pre}}$ and update $\theta$
     **end for**
   **end if**
   **for** $t = 1$ to 100,000 **do**
     Burn in walkers for $5N_e$ steps
     Sample configurations $\{\mathbf{R}\} \sim p_\theta$
     Estimate fidelity loss $L_{\mathrm{F}}$ and update $\theta$
   **end for**
   **for** $t = 1$ to 100 **do**
     Burn in walkers for $5N_e$ steps
     Sample configurations $\{\mathbf{R}\} \sim p_\theta$
     Evaluate fidelity $F$
   **end for**
   Record $F(N_e) \leftarrow \mathrm{mean}(F)$
   Store parameters $\theta_{\mathrm{last}} \leftarrow \theta$
**end for**
___

## 3. WF-Bench: A comprehensive dataset of important wavefunctions

We benchmark three classes of wavefunctions: topological states, superconducting wavefunctions, and Wigner crystal wavefunctions. Here, we introduce the form of wavefunctions.

### 3.1. Topological wavefunctions

We benchmark two famous trial wavefunctions for topological quantum states: Laughlin wavefunction(Laughlin, 1983) and Moore Read wavefunction.(Moore & Read, 1991) They corresponds to the ground states of the fractional quantum hall system at different filling factor $\nu = 1/m$. For odd $m$, the Laughlin wavefunction is

$$\Psi_{\mathrm{L}}(\{z_j\}) = \prod_{i<j}(z_i - z_j)^m \, \mathcal{G}(\{z_j\}), \qquad (1)$$

where $z_j = x_j + iy_j$, and $\ell_B = \sqrt{\hbar/(eB)}$ is the magnetic length. $\mathcal{G}(\{z_j\}) = \exp\left(-\sum_i |z_i|^2/4\ell_B^2\right)$ is the Gaussian

factor that originates from lowest Landau level (LLL) projection. Similarly, for even $m$, the Moore Read wavefunction is

$$\Psi_{\mathrm{MR}}(\{z_j\}) = \mathrm{Pf}\left(\frac{1}{z_i - z_j}\right) \prod_{i<j}(z_i - z_j)^m \, \mathcal{G}(\{z_j\}). \quad (2)$$

where $\mathrm{Pf}(\cdot)$ denotes the Pfaffian. As trial wavefunctions for the excited states of the fractional quantum hall problem, quasiholes can be added to the Laughlin and Moore Read states. For $N_h$ quasiholes at complex positions $\{\eta_a\}_{a=1}^{N_h}$, we write

$$\Psi_{X\mathrm{h}}(\{\mathbf{r}_j\}; \{\eta_a\}) = \left[\prod_{a=1}^{N_h}\prod_{i=1}^{N_e}(z_i - \eta_a)\right]\Psi_X(\{\mathbf{r}_j\}), \quad (3)$$

where $X \in \{\mathrm{L}, \mathrm{MR}\}$, and $\eta_a = \eta_{a,x} + i\eta_{a,y}$ denote the complex coordinates the quasiholes. Numerically, we place the quasiholes $\{\eta_a\}$ on a circle of radius $\sqrt{2mN_e}l_B/2$, i.e., half of the droplet radius. In our dataset, 12 topological states with different $m$ and $N_h$ are included. For example, `laughlin_m3h3` denotes a laughlin $m = 3$ state with 3 quasiholes. We show the amplitude and phase structures of topological states in Fig. 2a.

### 3.2. Superconducting wavefunctions

We benchmark the BCS family of wavefunctions with different pairing functions (De Gennes, 2018). The real space pairing function of the mean-field Cooper pair is given by

$$f(\mathbf{r}_i, \mathbf{r}_j) = \sum_{\mathbf{k}} g(\mathbf{k})\, e^{i\mathbf{k}\cdot(\mathbf{r}_i - \mathbf{r}_j)}, \quad (4)$$

where $\mathbf{r}_i$ and $\mathbf{r}_j$ denote the positions of two electrons within the cooper pair, $\mathbf{k}$ denotes the quantized momentum of the periodic system, and $g(\mathbf{k})$ is the Cooper-pair orbital, which takes different forms for different superconducting states. We show the shape of different pairing functions in Fig. 2c.

To impose particle-number conservation, we adopt the antisymmetrized geminal power (AGP) wavefunction,

$$\Psi_{\mathrm{BCS}}(\{\mathbf{r}_i, \sigma_i\}) = \mathcal{A}\big[\phi(1, 1')\phi(2, 2')\cdots\phi(N_p, N_p')\big], \quad (5)$$

where $\mathcal{A}$ denotes the antisymmetrizer, implemented as either $\mathrm{Pf}[\cdot]$ or $\det[\cdot]$, and $N_p = N_e/2$ is the number of Cooper pairs. Here, $\phi(\mathbf{r}_i, \mathbf{r}_j, \sigma_i, \sigma_j) = f(\mathbf{r}_i, \mathbf{r}_j)\langle\sigma_i, \sigma_j|\chi\rangle$ is the "geminal", constructed from the pairing function $f$, with $\sigma_i$ denoting the spin of the $i$-th electron and $|\chi\rangle$ the two-electron spin state, which is either singlet or triplet for BCS wavefunctions. In our dataset, we include 11 types of "geminal" functions, covering $s$-, $p$-, $d$-, and $f$-wave pairing with different spatial geometries and spin configurations. For example, `bcs_pp_tp` denotes a chiral $p_x + ip_y$ pairing with a triplet spin state $|\uparrow\uparrow\rangle$.

### 3.3. Wigner crystal wavefunctions

A Wigner crystal is a strongly correlated quantum state in which electrons spontaneously form crystalline patterns due to strong Coulomb interactions (Wigner, 1934). It corresponds to spontaneous breaking of continuous translational symmetry into a discrete lattice structure. A trial wavefunction for the Wigner crystal is constructed by first choosing a set of lattice sites $\{\mathbf{R}_i\}$. For a system with single-particle potential $V(\mathbf{r})$, the localized orbitals are taken as

$$\phi_j(\mathbf{r}_i) = \exp[-\alpha\, V(\mathbf{r}_i - \mathbf{R}_j)], \quad (6)$$

where $\alpha$ controls the degree of localization. We include three types of potentials: Gaussian orbital, squeezed Gaussian orbital, and a moiré orbital derived from perturbation theory(Luo et al., 2023). The Wigner crystal wavefunction is then given by

$$\Psi_{\mathrm{WC}}(\{\mathbf{r}_i\}) = \det[\phi_j(\mathbf{r}_i)]_{i,j=1}^{N_e}. \quad (7)$$

The lattice sites $\{\mathbf{R}_i\}$ are chosen according to the charge density patterns of the system. For the Wigner crystal wavefunctions included in our dataset, we consider $\{\mathbf{R}_i\} \in \{A, B, C\}$, where $A = [0, 0]$, $B = [\frac{1}{2}, \frac{1}{2}]$, and $C = [\frac{1}{3}, \frac{1}{3}]$. Here $[n, m]$ denotes the lattice coordinates within a hexagonal unit cell. For example, `wc_sgau_v1AB` denotes a $\nu = 2$ integer filling Wigner crystal state with a squeezed Gaussian orbital, and the lattice sites are $\{\mathbf{R}_i\} = \{A, B\}$. We show the charge density patterns of Wigner crystal wavefunctions in Fig. 2b.

## 4. Benchmarking Protocol

To match a neural network wavefunction $|\Psi_\theta\rangle$ with a target wavefunction $|\Phi\rangle$, a straightforward loss is based on the fidelity. We define the following:

$$L_{\mathrm{F}} = -\log\left[\frac{|\langle\Psi_\theta|\Phi\rangle|^2}{\langle\Psi_\theta|\Psi_\theta\rangle\langle\Phi|\Phi\rangle}\right] \quad (8)$$

Let $\Psi_\theta(\mathbf{R}) = \langle\mathbf{R}|\Psi_\theta\rangle$ (similarly for $\Phi(\mathbf{R})$), the gradient of $L_{\mathrm{F}}$ can be written in an expectation form by sampling from $p_\theta(\mathbf{R}) \propto |\Psi_\theta(\mathbf{R})|^2$:

$$\partial_\theta L_{\mathrm{F}} = 2\,\mathrm{Re}\left\{\mathbb{E}_{p_\theta}\left[(1 - \frac{\alpha(\mathbf{R})}{\mathbb{E}_{p_\theta}[\alpha(\mathbf{R})]})\,\partial_\theta \log \Psi_\theta^*(\mathbf{R})\right]\right\} \quad (9)$$

where $\alpha(\mathbf{R}) = \Phi(\mathbf{R})/\Psi_\theta(\mathbf{R})$. Despite its intuitive appeal, $L_{\mathrm{F}}$ becomes numerically challenging for large systems. In particular, $\alpha(\mathbf{R})$ demonstrates increasing variance as $N_e$ grows, due to interference. As a result, $L_{\mathrm{F}}$ suffers from a vanishing effective signal, making direct optimization unstable at large $N_e$.

For wavefunctions with moderate phase structure, the behavior of $\alpha(\mathbf{R})$ at the early stage of training can be controlled

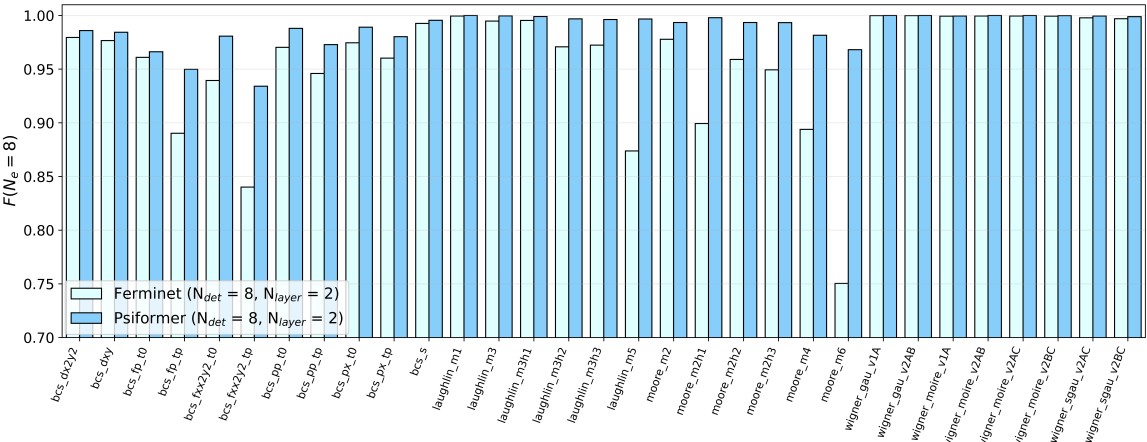

*Figure 3.* The value of $F(N_e = 8)$ for all 31 wavefunctions included in the dataset.

via transfer learning. However, for topological wavefunctions with complex phase windings, even transfer learning fails. One possible remedy is to match probability currents instead of fidelity , as proposed in (Nazaryan et al., 2025). However, we find the proposed protocol suffers from self-trapping during early stage of training. Since $|\Psi_\theta\rangle$ and $|\Phi\rangle$ are not normalized, the network can reduce the loss by matching amplitudes on a small set of configurations without moving probability mass globally, creating concentrated high-probability regions that freeze the walkers. To resolve this issue, we use the following loss :

$$L_{\text{pre}} = L_1 + \alpha L_2, \qquad (10)$$

where $\alpha$ is a hyper-parameter. The probability-matching term minimizes the difference between the probability densities of the two wavefunctions,

$$L_1 = \min_{c \in \mathbb{R}} \mathbb{E}_{p_{\text{mix}}}[(2\log[\alpha(\mathbf{R})] - c)^2], \qquad (11)$$

where $p_{\text{mix}} \propto \frac{p_\theta + p_t}{2}$. The current-matching term penalizes discrepancies in the per-particle phase gradients,

$$L_2 = \mathbb{E}_{p_\theta}\left[\sum_i \left\|\nabla_{\mathbf{R}_i}\psi_\theta(\mathbf{R}) - \nabla_{\mathbf{R}_i}\phi(\mathbf{R})\right\|^2\right]. \qquad (12)$$

An improvement of Eq. 10 over previous work is $L_1$, which is explicitly designed to mitigate self-trapping. Importantly, the constant $c$ effectively estimates the relative normalization factor between the two wavefunctions. By choosing $c = \mathbb{E}_{p_{\text{mix}}}[\log\alpha(\mathbf{R})]$, $L_1$ penalizes mismatches in probability density while remaining insensitive to overall normalization. The mixed sampling distribution $p_{\text{mix}}$ incorporates information from the target wavefunction into the training process, thereby preventing self-trapping.

For all loss functions used in this work, we use the kronecker-factored approximate curvature (KFAC) optimizer(Martens & Grosse, 2015) to obtain parameter update.

A detailed pseudo-code for the benchmarking protocol can be found in Alg. 1. We also discuss the relationship between observable error and fidelity in Appendix. D

## 5. Neural Network Wavefunction Architectures

We apply our dataset to two widely used NN wavefunction architectures, Ferminet (Pfau et al., 2020) and Psiformer (von Glehn et al., 2022). For both of the architectures, the input many body position $\mathbf{R}$ is first embedded as features. For Ferminet features includes both one electron and two electron part, where as only one electron feature is used for Psiformer.

The features are then passed into an expressive contextual encoder, which outputs a per electron tensor $h_{\text{orb}} \in \mathbb{R}^{N_e \times N_{\text{hid}}}$. For Ferminet, the contextual encoder includes a one body and two body streams, and $N_{\text{hid}}$ is defined as the width of the one body stream. As the original Ferminet concatenate the one body and two body streams, we use the SchNet(Schütt et al., 2018) enhanced version, where interstream mixing is achieved via continuous-filter convolutions. For Psiformer, the features are passed through multiple layers of self attention blocks + multilayer perceptron (MLP). Consequtively, and $N_{\text{hid}}$ refers to the attention dimension of the self attention block.

For both network, $h_{\text{orb}}$ is projected to real and imaginary part of $\phi_\theta^k$ via linear layers, where $k = 1, \ldots, N_{\text{det}}$. $\phi_\theta^k$ is also known as the neural orbitals. Finally, $\Psi_\theta(\mathbf{R})$ is generated by taking the sum of determinant of $\phi$:

$$\Psi_\theta(\mathbf{R}) = \sum_k^{N_{\text{det}}} \det[\phi_\theta^k] \qquad (13)$$

where $\phi_\theta^k \in \mathbb{C}^{N_e \times N_e}$ (We assume $\phi_\theta^k$ is complex for all

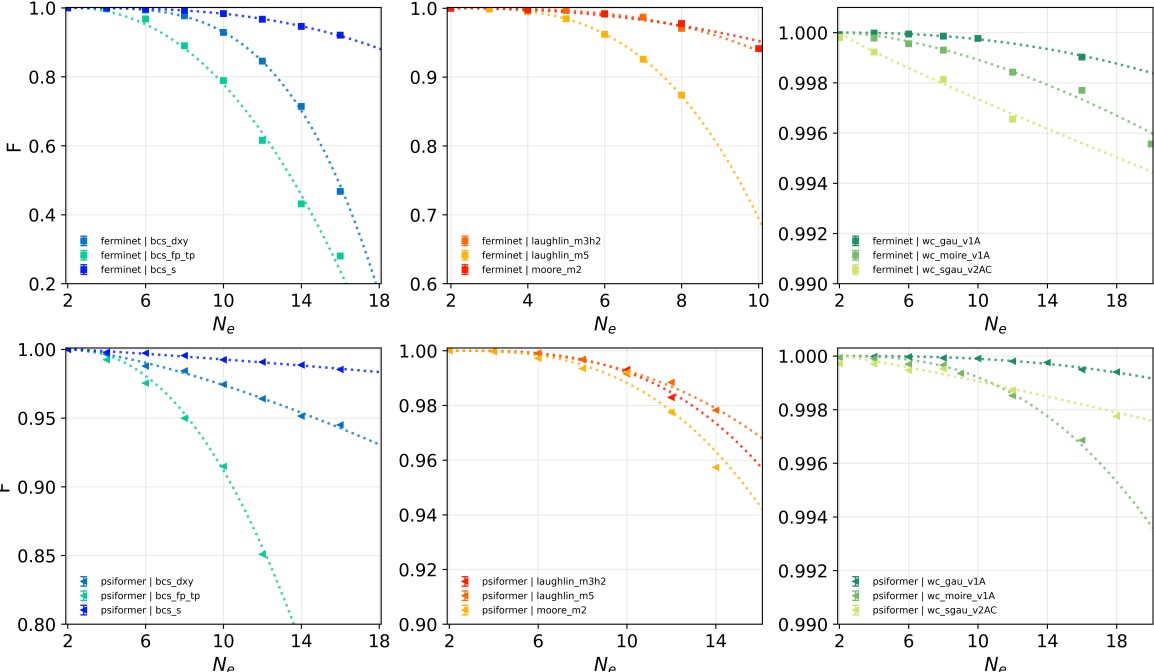

*Figure 4.* Fidelity scaling of 9 representative wavefunctions from superconductors (blue), topological states (red), and Wigner crystals (green). (a–c) Fidelity versus $N_e$ for Ferminet with $N_{\text{det}} = 8$ and $N_{\text{layer}} = 2$. (d–f) Corresponding results for Psiformer with $N_{\text{det}} = 8$ and $N_{\text{layer}} = 2$. Dotted lines indicate empirical fits of the form $F = 1 - \alpha(N_e - 2)^{\beta}$.

target wavefunctions for consistency). To impose physics prior to the orbitals, an envelope factor $\Omega_{i,j}^k$ can be included: $\phi_{i,j}^k \leftarrow \Omega_{i,j}\phi_{i,j}^k$. In particular, we choose an exponential decay envelope $\Omega_{i,j} = \exp(-\sigma_i|r_j|)$ for all topological wavefunctions to take account of strong decay in wavefunction imposed by the Gaussian factor $\mathcal{G}$. We refer readers to the original works cited above for further details on the architecture. The detailed parameters for both networks used in this work are included in the appendix.

## 6. Results and Discussions

By applying WF-Bench to Psiformer and Ferminet, we probe numerically the scaling behavior of the fidelity with respect to three key parameters: $N_e$, $N_{\text{layer}}$, and $N_{\text{det}}$.

### 6.1. Electron Number Scaling

We first present the scaling of fidelity with respect to $N_e$. We observe that the fidelity scaling of all wavefunctions can be well approximated by a power law: $F = 1 - \alpha(N_e - 2)^{\beta}$. The fitting parameter $\alpha$ sets an overall scale fidelity for small electron number, wile $\beta$ quantifies the rate at which fidelity decays with increasing $N_e$, reflecting the correlation strength and phase complexity of the target wavefunction. The constant shift by 2 is introduced assuming $F(N_e = 2)$ approaches unity, which is a reasonable assumption given nontrivial electron-electron correlations only emerge for

$N_e \geq 2$ in all wavefunctions in our dataset. In Fig. 3, we show the the $F(8)$ values across all 31 wavefunctions included in our datasets, spanning over topological states, superconductors and Wigner crystal. We found that across all wavefunctions included in our dataset, the performance of Psiformer exceeds Ferminet, despite relatively the same number of parameters (for topological states with $N_e = 10$, Ferminet has $7.3 \times 10^5$, whereas Psiformer has $8.3 \times 10^5$ parameters.)

We emphasize that representational difficulty is jointly determined by both the prefactor $\alpha$ and the exponent $\beta$: even when $\beta$ is small, a large $\alpha$ can induce substantial fidelity loss already at small system sizes. We therefore report $F(8)$ as a practical summary metric of representation power.

In Fig. 4, we select 9 representative wavefunctions from the three datasets, and plot the scaling of wavefunction fidelity $F$ with respect to the number of electrons. We observe that the fidelity scaling of all wavefunctions can be well approximated by a power law. The fitting parameters as well as the root mean square error (RMSE) of the fit can be found in Table. 1.

For superconductors, we include `bcs_s`, `bcs_dxy`, and `bcs_fp_tp`. The first two are time reversal symmetric singlet states, whereas the latter is a chiral triplet state with complex amplitudes and nontrivial phase structure. As shown in Fig. 4(a,d), the fidelity decays the fastest for the chiral

*Table 1.* Fidelity fittings of the 9 representative wavefunctions. For each wavefunction, we report the fidelity at $N_e = 8$, the fitted parameters $(\alpha, \beta)$, and the RMSE.

| NAME | FERMINET ($N_{\text{LAYER}} = 2$, $N_{\text{DET}} = 8$) | | | | PSIFORMER ($N_{\text{LAYER}} = 2$, $N_{\text{DET}} = 8$) | | | |
|---|---|---|---|---|---|---|---|---|
| | $F(8) \uparrow$ | $\alpha$ | $\beta$ | RMSE | $F(8) \uparrow$ | $\alpha$ | $\beta$ | RMSE |
| LAUGHLIN_M5 | 0.8737 | 5.66E-04 | 3.02 | 8.24E-04 | 0.9966 | 3.30E-05 | 2.60 | 1.04E-03 |
| MOORE_M2 | 0.9777 | 5.27E-04 | 2.17 | 5.90E-03 | 0.9934 | 3.39E-05 | 2.81 | 3.55E-03 |
| LAUGHLIN_M3H2 | 0.9707 | 1.24E-04 | 2.99 | 1.80E-03 | 0.9968 | 1.49E-05 | 3.01 | 9.88E-04 |
| BCS_S | 0.9925 | 5.38E-05 | 2.77 | 1.21E-03 | 0.9954 | 6.50E-04 | 1.17 | 3.62E-04 |
| BCS_FP_TP | 0.8902 | 2.25E-03 | 2.21 | 2.77E-02 | 0.9498 | 9.14E-04 | 2.20 | 8.97E-03 |
| BCS_DXY | 0.9765 | 4.31E-05 | 3.56 | 1.04E-02 | 0.9843 | 1.38E-03 | 1.41 | 1.73E-03 |
| WC_GAU_V1A | 0.9998 | 2.67E-06 | 2.21 | 3.41E-05 | 0.9999 | 5.06E-07 | 2.56 | 4.44E-05 |
| WC_MOIRE_V1A | 0.9993 | 3.79E-05 | 1.61 | 2.87E-04 | 0.9996 | 3.85E-06 | 2.56 | 1.10E-04 |
| WC_SGAU_V2AC | 0.9981 | 4.03E-04 | 0.91 | 1.52E-04 | 0.9995 | 8.57E-05 | 1.15 | 1.40E-04 |

triplet states in both Psiformer and Ferminet, which is an expected result of wavefunction complexity.

For topological states, we include laughlin_m3h2, laughlin_m5, and moore_m2. The first two are Laughlin wavefunctions with different $m$, while the latter is the Moore Read state. The laughlin_m3h2 wavefunction includes a quasihole excitation, making it more challenging to learn than the ground-state Laughlin wavefunction at $m = 3$. As shown in Fig. 4(b,e), the fidelity scaling of topological wavefunctions exhibits a large exponent in the power-law tendency: fidelity remain high at small $N_e$ but decay rapidly as $N_e$ increases. This behavior reflects the strong electron-electron correlations in topological states, whose complexity grows combinatorially with system size. Increasing $m$ further amplifies correlation effects, leading to a faster decay for laughlin_m5 compared with other topological targets. Although laughlin_m3h2 has a smaller $m$ than laughlin_m5, the Pfaffian structure of the Moore Read state makes moore_m2 more difficult to represent with determinant based neural ansätze, resulting in a lower $F(8)$. Finally, we observe that due to the highly nontrivial phase structure of topological states, network fidelity can collapse if the representation capacity is insufficient to capture enough phase information during pretraining, as illustrated in Fig. 3.

Lastly, for Wigner crystal states, we include wc_gau_v1A, wc_moire_v1A, and wc_sgau_v2AC. The first two correspond to $\nu = 1$ Wigner crystals with Gaussian and moiré orbitals, respectively, while the last represents a $\nu = 2$ Wigner crystal in a squeezed Gaussian orbital with $A$–$C$ hexagonal lattice stacking. Among the three classes of wavefunctions, Wigner crystal states exhibit the slowest fidelity decay with increasing electron number. This behavior arises because electrons are strongly localized by the Coulomb interactions, which suppresses complex phase winding compared with topological and superconducting states. This trend is reflected in the large $F(8)$ values reported in Table 1. Despite

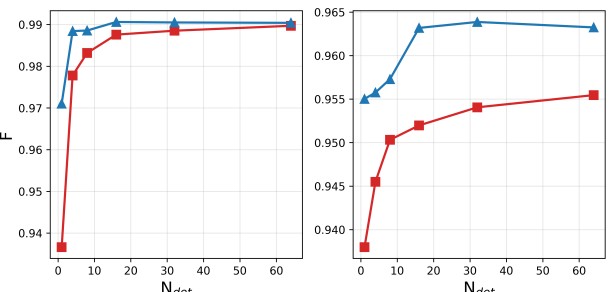

*Figure 5.* Fidelity scaling of bcs_s and moore_m2 with respect to $N_{\text{det}}$. Ferminet (red squares) is evaluated at $N_e = 10$, and Psiformer (blue triangles) is evaluated at $N_e = 14$. Both Psiformer and Ferminet are set as $N_{\text{layer}} = 2$.

the overall weak correlations, we observe a faster fidelity decay for wc_moire_v1A due to the increased structural complexity of the moiré orbital, as can be seen in Fig. 4(c,f).

### 6.2. Ablation study on $N_{\text{det}}$ and $N_{\text{layer}}$

In addition to fidelity scaling with respect to $N_e$, we also perform ablation study on the impact of key expressivity knobs, including the number of determinants $N_{\text{det}}$ and the number of layers $N_{\text{layer}}$. These are generic architectural parameters for NN wavefunctions based on neural network backflow formalism with NN orbitals.

In Fig. 5, we show the fidelity scaling of bcs_s and moore_m2 as a function of the number of determinants $N_{\text{det}}$. For small $N_{\text{det}}$, the fidelity increases rapidly, indicating a significant gain in representation power. However, once $N_{\text{det}}$ exceeds $\sim 32$, further increases yield only marginal improvements in fidelity. Since the computational cost scales linearly with $N_{\text{det}}$, using excessively large determinant expansions results in substantial computational overhead with limited performance improvement.

In Fig. 6, we present the fidelity scaling of bcs_s and

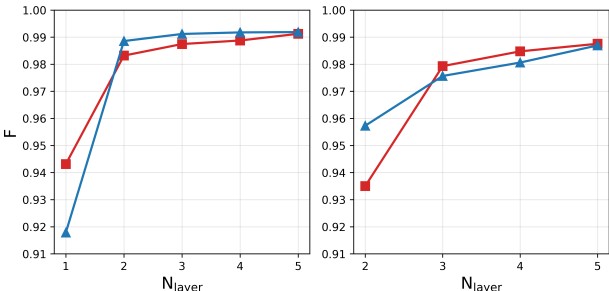

*Figure 6.* Fidelity scaling of `bcs_s` (a) and `moore_m2` (b) as a function of the number of layers $N_{\text{layer}}$. Ferminet (red squares) is evaluated at $N_e = 10$, and Psiformer (blue triangles) is evaluated at $N_e = 14$. Both Psiformer and Ferminet are set as: $N_{\text{det}} = 8$.

`moore_m2` as a function of the number of layers $N_{\text{layer}}$. When $N_{\text{layer}} = 1$, we observe the fidelity achieved by network is very low. In particular, for `moore_m2`, both Psiformer and Ferminet fail to represent the wavefunction. When $N_{\text{layer}}$ increases from 1 to 2, the fidelity improves markedly for both wavefunctions. Beyond $N_{\text{layer}} = 2$, further increases yield only modest gains in fidelity, indicating that deeper architectures do not substantially enhance representation power.

## 7. Conclusion

We introduce a comprehensive dataset, WF-Bench, for evaluating the representational power of neural network wavefunctions across physically distinct many body systems. The dataset spans three classes: topological states, superconductors, and Wigner crystals. It comprises 31 target wavefunctions covering a broad range of correlation structures, phase complexities, and anti-symmetrization mechanisms.

Applying this benchmark to two widely used architectures, FermiNet and PsiFormer, we systematically characterize how fidelity scales with system size, wavefunction class, and key architectural knobs. Across all targets, the achievable fidelity follows an empirical power law decay with increasing electron number. We further observe diminishing returns from commonly used architectural parameters: fidelity improves rapidly with the number of determinants at small values but saturates beyond a moderate threshold, while increasing network depth yields limited gains beyond a small number of layers.

Our comparison highlights clear differences in representational capacity between architectures: at comparable parameter counts, Psiformer consistently achieves higher fidelity than Ferminet across all target wavefunctions. More broadly, the dataset and benchmarking protocol provide a quantitative, architecture agnostic framework for evaluating neural network wavefunctions and can be readily applied to emerging models. In the meantime, we note that the optimization

protocols adopted in this work can still be further improved, which could potentially affect the scaling behavior established in this work. We leave such investigations to future works.

We expect this framework to serve as a reference point for the design, benchmarking and theoretical analysis for new NN wavefunction architectures. We also envision this dataset as a community driven resource that can be extended to additional classes of wavefunctions, including lattice models under second quantized formalism, chemical molecules, and other strongly correlated quantum systems.

## Acknowledgement

DL acknowledges support from Beijing Municipal Science and Technology Commission and Zhongguancun Science Park Administrative Committee (No. 20251090054).

## Data Availability

A `JAX` implementation of target wavefunctions and loss functions is available at `https://github.com/L0bsterkun/WF-Bench`.

## Impact Statement

This work presents a comprehensive dataset and benchmark protocol for representability scaling of neural network wavefunctions. There are many potential societal consequences of our work, none of which we feel must be specifically highlighted here.

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

# A. Details on the WF-Bench

The WF-Bench dataset includes a total of 31 wavefunctions. The $F(8)$ value, as well as the fitting parameters and fitting RMSE of each wavefunctions is included in Table. 6.

## A.1. Topological wavefunctions

For topological wavefunctions, we include 12 wavefunctions. The setting of each wavefunctions can be found in Table. 2. We note that during the training for topological wavefunctions, we apply a pre-determinant exponential envelope to take account of the Gaussian factor $\mathcal{G}(\{z_j\})$ in the target wavefunction. The Gaussian factor originates from the lowest Landau level (LLL) projection, which ensures the normalizabiltity of the wavefunctions. We also acknowledge that alternative boundary conditions have also been used, for examples a torus.

*Table 2.* Parameters for the topological wavafunctions.

| Name | m | $N_h$ | Antisymmetrizer |
|---|---|---|---|
| laughlin_m1 | 1 | 0 | - |
| laughlin_m3 | 3 | 0 | - |
| laughlin_m5 | 5 | 0 | - |
| laughlin_m3h1 | 3 | 1 | - |
| laughlin_m3h2 | 3 | 2 | - |
| laughlin_m3h3 | 3 | 3 | - |
| moore_m2 | 2 | 0 | Pfaffian |
| moore_m4 | 4 | 0 | Pfaffian |
| moore_m6 | 6 | 0 | Pfaffian |
| moore_m2h1 | 2 | 1 | Pfaffian |
| moore_m2h2 | 2 | 2 | Pfaffian |
| moore_m2h3 | 2 | 3 | Pfaffian |

## A.2. Superconducting wavefunctions

For superconducting wavefunctions, we start by choosing a single-particle dispersion $\epsilon(\mathbf{k})$. The mean-field Cooper pair orbital is defined as:(De Gennes, 2018)

$$g(\mathbf{k}) \equiv \frac{v_{\mathbf{k}}}{u_{\mathbf{k}}} = \frac{\Delta(\mathbf{k})}{\epsilon(\mathbf{k}) + E(\mathbf{k})}, \tag{14}$$

where $E(\mathbf{k}) = \sqrt{\epsilon(\mathbf{k})^2 + |\Delta(\mathbf{k})|^2}$ is the Bogoliubov quasiparticle dispersion. We use a parabolic dispersion $\epsilon(\mathbf{k}) = \frac{1}{2}(|\mathbf{k}|^2 - k_F^2)$ and choose $k_F = \sqrt{\frac{2\pi N_e}{A}}$ for area $A$. We choose $A$ such that $k_F$ is fixed at $\sqrt{\frac{\pi}{2}}$ across different $N_e$. The $k-$ space pairing orbitals are then Fourier transformed to the pairing function we used in the antisymmetrized geminimal power (AGP) function, as in Eq. 5. For a periodic square box with length $L$, the momentum $\mathbf{k} = \frac{2\pi}{L}(n_x, n_y)$ is quantized with integers $n_x, n_y \in \mathbb{Z}$. Numerically, we include all momentum with $|\mathbf{k}| < 3k_F$ to ensure convergence. The detailed setting of each superconducting wavefunctions can be found in Table. 3.

## A.3. Wigner crystal wavefunctions

We implement the Wigner crystal wavefunctions by choosing a form of orbital function. The Gaussian and squeezed Gaussian orbital included in our dataset takes the form of:

$$\phi_j^{\text{gau}}(\mathbf{r}_i) = \exp[-\alpha \tilde{\mathbf{r}}_{i,j}^2], \qquad \phi_j^{\text{sgau}}(\mathbf{r}_i) = \exp\left[-\alpha \left(\frac{(\tilde{r}_{x;\,i,j})^2}{s^2} + (\tilde{r}_{y;\,i,j})^2\, s^2\right)\right], \tag{15}$$

where $\tilde{\mathbf{r}}_{i,j}$ is the minimal image distance between $\mathbf{r}_i$ and $\mathbf{R}_j$. We choose $s = 0.5$ for $\phi_j^{\text{sgau}}(\mathbf{r}_i)$.

For the moiré orbital, we choose:

$$\phi_j^{\text{moiré}}(\mathbf{r}_i) = \exp[-\alpha(V(r_{ij}) - V(0))] \tag{16}$$

*Table 3.* Parameters for the superconducting wavefunctions.

| Name | $\Delta_{\mathbf{k}}$ | $|\chi\rangle$ | Antisymmetrizer |
|------|------------------------|----------------|-----------------|
| bcs_s | $\Delta_0$ | $\frac{1}{\sqrt{2}}(|\uparrow\downarrow\rangle - |\downarrow\uparrow\rangle)$ | Determinant |
| bcs_dxy | $\Delta_0 \sin k_x \sin k_y$ | $\frac{1}{\sqrt{2}}(|\uparrow\downarrow\rangle - |\downarrow\uparrow\rangle)$ | Determinant |
| bcs_dx2y2 | $\Delta_0(\cos k_x - \cos k_y)$ | $\frac{1}{\sqrt{2}}(|\uparrow\downarrow\rangle - |\downarrow\uparrow\rangle)$ | Determinant |
| bcs_px_t0 | $\Delta_0 \sin k_x$ | $\frac{1}{\sqrt{2}}(|\uparrow\downarrow\rangle + |\downarrow\uparrow\rangle)$ | Pfaffian |
| bcs_pp_t0 | $\Delta_0(\sin k_x + i \sin k_y)$ | $\frac{1}{\sqrt{2}}(|\uparrow\downarrow\rangle + |\downarrow\uparrow\rangle)$ | Pfaffian |
| bcs_px_tp | $\Delta_0 \sin k_x$ | $|\uparrow\uparrow\rangle$ | Pfaffian |
| bcs_pp_tp | $\Delta_0(\sin k_x + i \sin k_y)$ | $|\uparrow\uparrow\rangle$ | Pfaffian |
| bcs_fxx2y2_tp | $\Delta_0 \sin k_x(\cos k_x - \cos k_y)$ | $|\uparrow\uparrow\rangle$ | Pfaffian |
| bcs_fxx2y2_t0 | $\Delta_0 \sin k_x(\cos k_x - \cos k_y)$ | $\frac{1}{\sqrt{2}}(|\uparrow\downarrow\rangle + |\downarrow\uparrow\rangle)$ | Pfaffian |
| bcs_fp_tp | $\Delta_0(\sin k_x + i \cos k_y)^3$ | $|\uparrow\uparrow\rangle$ | Pfaffian |
| bcs_fp_t0 | $\Delta_0(\sin k_x + i \cos k_y)^3$ | $\frac{1}{\sqrt{2}}(|\uparrow\downarrow\rangle + |\downarrow\uparrow\rangle)$ | Pfaffian |

where $V$ is the moiré potential:

$$
\begin{aligned}
V_{\mathrm{moire}}(r, \theta) =\ & -6\cos\phi + 8\pi^2 \cos\phi\, r^2 + \frac{16\pi^3}{3\sqrt{3}} \sin\phi\,\sin(3\theta)\, r^3 \\
& - \frac{8\pi^4}{3} \cos\phi\, r^4 - \frac{16\pi^5}{9\sqrt{3}} \sin\phi\,\sin(3\theta)\, r^5 + \frac{16\pi^6}{405} \cos\phi\left(10 - \cos(6\theta)\right) r^6.
\end{aligned}
\tag{17}
$$

We choose $\alpha = 10$ for all 8 wavefunctions included. The lattice vector is choosen as $\mathbf{a}_1 = n_x\,(1,0)$, $\mathbf{a}_2 = n_y\left(\frac{1}{2}, \frac{\sqrt{3}}{2}\right)$, where $n_x$ and $n_y$ is the number of unit cell included in the super lattice.

## B. Details on Network and Optimization Parameters

Here, we introduce details on the network as well as optimization parameter used during the training process. The network parameters used is listed in Table. 4. The KFAC optimizer's parameter used is listed in Table. 5

## C. Details on the Loss Functions

Here, we detail the loss functions used in the benchmarking protocol.

### C.1. Fidelity loss

. By choosing $L_F = -\log F = -\log[\frac{|\langle\Psi_\theta|\Phi\rangle|^2}{\langle\Psi_\theta|\Psi_\theta\rangle\langle\Phi|\Phi\rangle}]$, we have:

$$
L_{\mathrm{fid}} = \log[\sum_{\mathbf{R}} |\Psi_\theta(\mathbf{R})|^2] + \log[\sum_{\mathbf{R}} |\Phi(\mathbf{R})|^2] - \log[|\sum_{\mathbf{R}} \Psi_\theta^*(\mathbf{R})\Phi(\mathbf{R})|^2]
\tag{18}
$$

*Table 4.* Network architecture parameters.

| | Psiformer | Ferminet |
|------|-----------|----------|
| attention dimension | 64 | – |
| number of attention head | 4 | – |
| MLP hidden dims (1e) | 256 | 512 |
| MLP hidden dims (2e) | – | 32 |
| use layer norm | True | – |
| user last layer | – | True |
| separate spin channel | – | False |
| electron convolution dimension | – | 16 |
| complex output | True | True |

*Table 5.* Optimizer parameters. The learn rate is chosen as $lr = \alpha(\frac{1}{1+\frac{t}{\tau}})$, where $\tau = 1 \times 10^{-4}$. $\alpha$ is set as $1 \times 10^{-3}$ for the first wavefunction in the transfer series, and $1 \times 10^{-4}$ for the rest of the wavefunctions.

| Name | KFAC |
|---|---|
| batch size | 4096 |
| damping damping | $1 \times 10^{-4}$ |
| minimal damping | $1 \times 10^{-6}$ |
| norm constraint | $1 \times 10^{-3}$ |
| Momentum | 0.0 |
| Curvature EMA | 0.95 |
| Estimation mode | fisher_exact |

Its gradient is then :

$$
\begin{aligned}
\partial_\theta L_{\text{fid}} &= \partial_\theta\Big[\log[\sum_{\mathbf{R}} |\Psi_\theta(\mathbf{R})|^2]\Big] - \partial_\theta\Big[\log[|\sum_{\mathbf{R}} \Psi_\theta^*(\mathbf{R})\Phi(\mathbf{R})|^2]\Big] \\
&= \partial_\theta\Big[\log[\sum_{\mathbf{R}} |\Psi_\theta(\mathbf{R})|^2]\Big] - 2\text{Re}\Big\{\partial_\theta\Big[\log[\sum_{\mathbf{R}} \Psi_\theta^*(\mathbf{R})\Phi(\mathbf{R})]\Big]\Big\}
\end{aligned} \tag{19}
$$

For the first part, we have:

$$
\partial_\theta\Big[\log[\sum_{\mathbf{R}} |\Psi_\theta(\mathbf{R})|^2]\Big] = \frac{2N_\theta \text{Re}\Big\{\mathbb{E}_{p_\theta}\Big[\partial_\theta\log[\Psi_\theta^*(\mathbf{R})]\Big]\Big\}}{N_\theta} = 2\text{Re}\Big\{\mathbb{E}_{p_\theta}\Big[\partial_\theta[\log[\Psi_\theta^*(\mathbf{R})]]\Big]\Big\} \tag{20}
$$

For the second part, we have:

$$
-2\text{Re}\Big\{\partial_\theta\Big[\log[\sum_{\mathbf{R}} \Psi_\theta^*(\mathbf{R})\Phi(\mathbf{R})]\Big]\Big\} = -2\text{Re}\Big\{\frac{\partial_\theta[\sum_{\mathbf{R}} \Psi_\theta^*(\mathbf{R})\Phi(\mathbf{R})]}{\sum_{\mathbf{R}} \Psi_\theta^*(\mathbf{R})\Phi(\mathbf{R})}\Big\} = -2\text{Re}\Big\{\frac{\mathbb{E}_{p_\theta}\Big[[\Phi(\mathbf{R})/\Psi_\theta(\mathbf{R})] \cdot \partial_\theta\log[\Psi_\theta^*(\mathbf{R})]\Big]}{\mathbb{E}_{p_\theta}[\Phi(\mathbf{R})/\Psi_\theta(\mathbf{R})]}\Big\}
$$

$$\tag{21}$$

Therefore, we have:

$$
\partial_\theta L_{\text{fid}} = 2\text{Re}\Big\{\mathbb{E}_{p_\theta}\Big[[1 - \frac{\alpha(\mathbf{R})}{\mathbb{E}_{p_\theta}[\alpha(\mathbf{R})]}] \cdot \partial_\theta\log[\Psi_\theta^*(\mathbf{R})]\Big]\Big\} \tag{22}
$$

where $\alpha(\mathbf{R}) \equiv \Phi(\mathbf{R})/\Psi_\theta(\mathbf{R})$ is the complex ratio of the target and network wavefunction. We note that an alternative loss function $L = 1 - F$ is also widely used. Nevertheless, as $-\frac{\partial \log F}{\partial \theta} = -\frac{1}{F}\frac{\partial F}{\partial \theta}$, the two loss functions converge to the same value as $F \to 1$ and the constant factor can be absorbed into the learning rate.

## C.2. Mixed sampling Probability Loss

The pretraining loss is defined as

$$
L_{\text{pre}} = L_1 + \alpha L_2, \tag{23}
$$

with $\alpha = 1 - e^{-\tau/t}$. Numerically, we choose $\tau = 2 \times 10^4$ for the first wavefunction in the transfer series, and $\tau = 1 \times 10^3$ for the rest of the wavefunctions. As the form of $L_2$ is already shown in the main-body of the paper, here, we detail the derivations for $L_1$. We start from the symmetric Kullback–Leibler (KL) divergence, and define our loss function as:

$$
\begin{aligned}
L &= \frac{1}{2}[\text{KL}(p_\theta||p_\text{t}) + \text{KL}(p_\text{t}||p_\theta)] \\
&= \frac{1}{2}[\mathbb{E}_{p_0}(\ln p_\theta - \ln p_\text{t}) + \mathbb{E}_{p_\text{t}}(\ln p_\text{t} - \ln p_\theta)]
\end{aligned} \tag{24}
$$

However, as $|\Psi_\theta\rangle$ and $|\Phi\rangle$ are not normalized, $p_\theta$ and $p_\text{t}$ are unknown. We define $\overline{\overline{\overline{}}}\ln(\frac{N_\text{t}}{N_\theta})$, where $N_\text{t}$ and $N_\theta$ is the normalization factor of $|\Phi\rangle$ and $|\Psi_\theta\rangle$, respectively. The loss function can be rewritten as:

$$
L = \frac{1}{2}\Big[\mathbb{E}_{p_\theta}\big[(\ln|\Psi_\theta(\mathbf{R})|^2 - \ln|\Phi(\mathbf{R})|^2 - c)^2\big] + \mathbb{E}_{p_\text{t}}\big[(\ln|\Phi(\mathbf{R})|^2 - \ln|\Psi_\theta(\mathbf{R})|^2 + c)^2\big]\Big] \tag{25}
$$

*Table 6.* Fidelity fittings of the 31 wavefunctions in WF-Bench. For each wavefunction, we report the fidelity at $N_e = 8$, the fitted parameters $(\alpha, \beta)$, and the root-mean-square error (RMSE).

| NAME | **FERMINET** ($N_{\text{LAYER}} = 2,\ N_{\text{DET}} = 8$) | | | | **PSIFORMER** ($N_{\text{LAYER}} = 2,\ N_{\text{DET}} = 8$) | | | |
|---|---|---|---|---|---|---|---|---|
| | $F(8)$ | $\alpha$ | $\beta$ | RMSE | $F(8)$ | $\alpha$ | $\beta$ | RMSE |
| LAUGHLIN_M1 | 0.9994 | 1.11E-05 | 2.13 | 3.21E-04 | 0.9999 | 3.40E-07 | 3.16 | 1.25E-04 |
| LAUGHLIN_M3 | 0.9947 | 4.68E-05 | 2.67 | 2.21E-04 | 0.9996 | 7.26E-07 | 3.31 | 5.21E-05 |
| LAUGHLIN_M5 | 0.8737 | 5.66E-04 | 3.02 | 8.24E-04 | 0.9966 | 3.30E-05 | 2.60 | 1.04E-03 |
| LAUGHLIN_M3H1 | 0.9954 | 3.50E-06 | 4.06 | 4.17E-04 | 0.9989 | 1.28E-05 | 2.43 | 3.13E-04 |
| LAUGHLIN_M3H2 | 0.9707 | 1.24E-04 | 2.99 | 1.80E-03 | 0.9968 | 1.49E-05 | 3.01 | 9.88E-04 |
| LAUGHLIN_M3H3 | 0.9722 | 1.92E-04 | 2.72 | 1.79E-03 | 0.9961 | 3.92E-04 | 1.29 | 2.54E-04 |
| MOORE_M2 | 0.9777 | 5.27E-04 | 2.17 | 5.90E-03 | 0.9934 | 3.39E-05 | 2.81 | 3.55E-03 |
| MOORE_M4 | 0.8937 | 9.84E-05 | 3.95 | 6.62E-03 | 0.9815 | 1.23E-04 | 2.75 | 7.97E-03 |
| MOORE_M6 | 0.7068 | 2.00E-03 | 2.76 | 7.69E-03 | 0.9681 | 2.74E-04 | 2.64 | 6.97E-03 |
| MOORE_M2H1 | 0.8993 | 1.70E-04 | 3.52 | 8.40E-03 | 0.9978 | 7.12E-07 | 4.67 | 1.40E-03 |
| MOORE_M2H2 | 0.9590 | 4.22E-04 | 2.49 | 2.95E-03 | 0.9934 | 2.26E-05 | 3.11 | 1.45E-03 |
| MOORE_M2H3 | 0.9493 | 7.10E-04 | 2.22 | 8.23E-03 | 0.9933 | 5.93E-05 | 2.62 | 1.75E-03 |
| BCS_S | 0.9925 | 5.38E-05 | 2.77 | 1.21E-03 | 0.9954 | 6.50E-04 | 1.17 | 3.62E-04 |
| BCS_DXY | 0.9765 | 4.31E-05 | 3.56 | 1.04E-02 | 0.9843 | 1.38E-03 | 1.41 | 1.73E-03 |
| BCS_DX2Y2 | 0.9795 | 2.47E-05 | 3.68 | 8.89E-03 | 0.9859 | 2.89E-04 | 2.10 | 3.69E-03 |
| BCS_PX_T0 | 0.9744 | 1.77E-04 | 2.65 | 3.83E-03 | 0.9890 | 9.23E-04 | 1.29 | 1.22E-03 |
| BCS_PP_T0 | 0.9703 | 2.32E-04 | 2.67 | 1.43E-03 | 0.9880 | 3.99E-04 | 1.82 | 8.49E-04 |
| BCS_PX_TP | 0.9602 | 4.77E-04 | 2.55 | 2.49E-03 | 0.9802 | 9.59E-04 | 1.69 | 1.26E-03 |
| BCS_PP_TP | 0.9459 | 9.13E-04 | 2.36 | 8.59E-03 | 0.9728 | 5.04E-04 | 2.17 | 7.11E-04 |
| BCS_FXX2Y2_TP | 0.8401 | 1.34E-02 | 1.58 | 2.55E-02 | 0.9340 | 2.54E-03 | 1.83 | 8.32E-03 |
| BCS_FXX2Y2_T0 | 0.9394 | 3.72E-04 | 2.84 | 1.41E-02 | 0.9806 | 2.43E-03 | 1.18 | 1.65E-03 |
| BCS_FP_TP | 0.8902 | 2.25E-03 | 2.21 | 2.77E-02 | 0.9498 | 9.14E-04 | 2.20 | 8.97E-03 |
| BCS_FP_T0 | 0.9609 | 1.02E-04 | 3.31 | 2.76E-02 | 0.9661 | 1.19E-03 | 1.81 | 3.85E-03 |
| WC_GAU_V1A | 0.9998 | 2.67E-06 | 2.21 | 3.41E-05 | 0.9999 | 5.06E-07 | 2.56 | 4.44E-05 |
| WC_GAU_V2AB | 0.9998 | 1.49E-06 | 2.38 | 6.27E-05 | 0.9999 | 4.63E-07 | 2.55 | 7.21E-05 |
| WC_SGAU_V2AC | 0.9981 | 4.03E-04 | 0.91 | 1.52E-04 | 0.9995 | 8.57E-05 | 1.15 | 1.40E-04 |
| WC_SGAU_V2BC | 0.9969 | 2.34E-04 | 1.45 | 1.36E-03 | 0.9988 | 5.19E-05 | 1.71 | 8.60E-04 |
| WC_MOIRE_V1A | 0.9993 | 3.79E-05 | 1.61 | 2.87E-04 | 0.9996 | 3.85E-06 | 2.56 | 1.10E-04 |
| WC_MOIRE_V2AB | 0.9994 | 4.84E-06 | 2.60 | 6.26E-04 | 0.9998 | 1.13E-05 | 1.64 | 4.89E-04 |
| WC_MOIRE_V2AC | 0.9994 | 1.75E-06 | 3.15 | 1.61E-03 | 0.9999 | 1.21E-05 | 1.48 | 4.86E-04 |
| WC_MOIRE_V2BC | 0.9993 | 4.77E-04 | 0.61 | 1.01E-03 | 0.9998 | 2.46E-04 | 0.47 | 1.08E-03 |

Using quadratic KL loss instead of linear gives:

$$L_1 = \frac{1}{2}\left[ \mathbb{E}_{p_\theta}\left[(2\alpha(\mathbf{R}) - c)^2\right] + \mathbb{E}_{p_t}\left[(2\alpha(\mathbf{R}) + c)^2\right] \right] \tag{26}$$

Then we have:

$$\frac{\partial L}{\partial c} = \frac{1}{2}\left[ 2(\mathbb{E}_{p_\theta}[2\alpha(\mathbf{R})] + \mathbb{E}_{p_t}[2\alpha(\mathbf{R})]) + 4c \right] \tag{27}$$

Setting $\frac{\partial L}{\partial c} = 0$ gives $c^* = -\frac{1}{2}(\mathbb{E}_{p_\theta}[2\alpha(\mathbf{R})] + \mathbb{E}_{p_t}[2\alpha(\mathbf{R})])$, which is a numerically stable expectation variable in logarithmic space. Plugging $c^*$ back to Eq. 26 gives :

$$L_1 = \mathbb{E}_{p_{\text{mix}}}\left[(2\alpha(\mathbf{R}) - c^*)^2\right] \tag{28}$$

Moreover, as $\frac{\partial L}{\partial c}\big|_{c=c^*} = 0$:

$$\frac{d}{d\theta}L_1 = \frac{\partial L_1}{\partial \theta} + \frac{\partial L_1}{\partial c}\frac{dc}{d\theta} = \frac{\partial L_1}{\partial \theta} \tag{29}$$

which means $c$ can be treated as a constant during the backpropagation.

## D. Relation between Fidelity and Observable error

As fidelity is used as the benchmark metric for wavefunction expressivity, it is necessary to discuss the gap between fidelity and errors in observables, which are physically important and relevant in different training scenarios (for example, variational Monte Carlo methods). In general, the following bound exists for any bounded operator $\hat{O}$ and $F = |\langle\psi|\phi\rangle|^2$:

$$|\langle\psi|\hat{O}|\psi\rangle - \langle\phi|\hat{O}|\phi\rangle| \leq 2||\hat{O}||\sqrt{1-F} \tag{30}$$

Therefore, the error in the expectation value of any operator with bounded norm is bounded by infidelity. We would also like to point out the observable error scaling law could be an interesting direction for future works.

