# OpenReview forum: "WF-Bench: A Benchmark for Neural-Network WaveFunction Expressivity and Scaling Laws"
_ICML.cc/2026/Conference — ICML 2026 regular_

### Official Review · Reviewer_o4mZ · 2026-02-28

**Soundness:** 3
**Presentation:** 3
**Significance:** 3
**Originality:** 3
**Overall Recommendation:** 5
**Confidence:** 4

**Summary:**

The paper proposes a set of benchmark problems that tackle the approximation of the wave function using neural networks as applied in computational quantum physics. A unified training and evaluation protocol is proposed across different NN wavefunction architectures and expressiveness parameters. This finally leads to recovering empirical scaling laws by systematically varying electron number
and network capacity.
These results are validated on two methods Psiformer and Ferminet with across all 31 wavefunctions.

**Compliance With Llm Reviewing Policy:**

Affirmed.

**Final Justification:**

My final recommendation is to accept and after the answers and explanations from the authors, my confidence has increased to 4.

**Key Questions For Authors:**

1- Why did you choose these three classes of wavefunctions specifically?
2- At the beginning of Section 6, you say "We observe that the fidelity scaling of all wavefunctions can be well approximated by a power law". Where do you observe that exactly? Could you point out in your results where we can see that?
3- Can you give some insights why Ferminet fails specifially to represent moore m2 at Ne = 8, and not others?
4- You say that "the performance of Psiformer exceeds Ferminet" because of F but it seems like the RMSE is lower for Ferminet for the cases you reported. Could you discuss this in more details?
5- It seems you only obtain a scaling law for Ne. Is that the case? If yes, then the way you introduce scaling laws is a bit misleading and too general. This should be made more specific.

**Limitations:**

Limitations were not really discussed. For example, what methods are not covered?

**Strengths And Weaknesses:**

*Stengths*

Soundess: The claims are overall well supported numerically with examples and applications

Presentation: The paper is well written.

Significance: The problem of evaluating neural network wave function is important because it provides a highly expressive that can efficiently approximate strongly correlated quantum many-body states and having a benchmark to evaluate the methods is very valuable for the community.

Originality: As far as I know, this is original work.

*Weaknesses*

Soundess: Some of the choices for the methods, metrics, and special wave functions could be better documented.

Presentation: In the numerical section, the results could be better discussed. You could move the figures that take a lot of space to the appendix to have more space.

Significance: /

Originality: /

---

> ### Author Rebuttal · Authors · 2026-03-31
>
> We are grateful for the reviewers on the validation of the originality, quality, and presentation of this work. We have also included additional experimental data in the same anonymous GitHub link for reviewer’s verification. We have made targeted changes and new contents to reflect the weaknesses(W) and questions(Q) raised by the reviewer:
>
> W1. Soundness
>
> We thank the reviewer for the valuable suggestion. We have added more details on the training procedures, loss functions and reported fidelity in the revised manuscript. We have also added more details on the detailed parameters of wavefunctions in the appendix.
>
> W2. Presentation.
>
> We thank the reviewer for the valuable suggestion. We have added discussions on the fidelity scaling on different wavefunctions, as well as the performance of each neural network architectures in the revised manuscript.
>
> Q1. We include target wavefunctions from three major classes—topological states, superconducting states, and Wigner crystals—covering a broad range of physically important systems with intrinsically different correlation structures. The topological class includes well-known trial wavefunctions that originated from the fractional quantum Hall systems projected to lowest Landau-Levels. They are found in many topological materials such as rhombohedral graphene and transition metal dichalcogenides. The superconducting class includes BCS–type wavefunctions, which are found in many superconducting materials. Two famous examples would be cuprate and nickelate. The Wigner-crystal class includes states that exhibit spontaneous symmetry breaking in contrast to conventional Fermi gas, which have be seen in quantum materials where interaction energy dominates kinetic energy.
>
> Q2. We thank the reviewer for this comment. In Figure 4, we show the fidelity scaling for 9 representative wavefunctions drawn from the three datasets and fit the results using a power-law function (dotted lines). Indeed, we find that, across all wavefunctions in our dataset, the fidelity achieved by NN wavefunctions can be approximated by a power-law scaling on the number of electrons.
> Q3. Moore_m6 is a highly frustrated wavefunction with complex phase structures and windings due to the large value of m. Ferminet also has weaker expressive power compared with psiformer. During the pretrain stage, Ferminet’s expressivity is insufficient to capture enough phase structure, and the following fidelity training has a trivial signal and fails converging to a meaningful value.
>
> Q4. The value of RMSE reported in the table represents the quality of the power law fitting. However, a direct evidence of expressivity performance is the fidelity value at $N_e = 8$ ($F(8)$ value in Table. I). We found that Ferminet generally delivers a lower $F(8)$ value than Psiformer across all target wavefunctions included in our dataset.
>
> Q5.  The scaling in $N_e$ is important to the physics in the thermodynamic limit. Determining the scaling of NN wavefunctions with $N_e$ is therefore important to probe their potential to be extended to the thermodynamic limit. In addition to the scaling in $N_e$, we have also included ablation study on how the fidelity scales with $N_{\rm layer}$ and $N_{\rm det}$, as shown in Figure 5 and 6 in the manuscript. We have revised the manuscript to be more specific.
>
> We thank the reviewer for the detailed review. We hope these clarifications adequately address the reviewer's concerns. We would truly appreciate it if the reviewer would consider raising the score. We are happy to answer any further questions during the discussion period.

---

> > ### Author Rebuttal · Reviewer_o4mZ · 2026-04-02
> >
> > Thank you for the answers and the work. I would like to keep my score.

---

> > > ### Author Response · Authors · 2026-04-06
> > >
> > > We sincerely appreciate the reviewer’s positive assessment of our work and their thoughtful feedback. We also thank the reviewer for the time and effort spent in the review process and the follow-up discussions, which has significantly helped us to further improve the quality and clarity of the manuscript. We would truly appreciate it if the additional results and revisions could help increase the reviewer’s confidence in the evaluation.

---

### Official Review · Reviewer_jUjv · 2026-03-12

**Soundness:** 3
**Presentation:** 3
**Significance:** 3
**Originality:** 2
**Overall Recommendation:** 4
**Confidence:** 3

**Summary:**

This paper introduces a dataset and protocol for evaluating neural network wavefunction expressivity. By matching networks to diverse target states, the authors empirically study how fidelity scales with electron count, determinant number, and network depth for FermiNet and PsiFormer.

**Compliance With Llm Reviewing Policy:**

Affirmed.

**Final Justification:**

All my questions have been resolved.

**Key Questions For Authors:**

How do you theoretically justify the empirical power-law fit $F = 1 - \alpha(N_e - 2)^\beta$ given that the overlap between many-body states fundamentally exhibits exponential decay with $N_e$?

Have you evaluated intensive/size-extensive metrics, such as fidelity per particle or energy density equivalents, to better characterize the scaling behavior toward the thermodynamic limit?

Can you provide empirical evidence (e.g., tracking gradient variance across layers during optimization) to prove that the performance saturation with increasing $N_{layer}$ is genuinely an expressivity limit and not a failure of the KFAC optimizer in deeper landscapes?

How does the proposed $L_{pre}$ loss function, specifically the dynamic calculation of $c^*$, maintain stability when Monte Carlo variance inevitably blows up at larger $N_e$?

**Limitations:**

yes

**Strengths And Weaknesses:**

Strengths:

- The paper is well-structured, with a clear and logical separation between the physical dataset descriptions, the optimization protocol, and the empirical results.
- Providing the specific mathematical forms for all 31 wavefunctions along with the exact KFAC optimization hyperparameters significantly aids the reproducibility of the benchmark.

Weaknesses:

- The empirical scaling laws are extrapolated from extremely small systems (maximum $N_e \approx 18$). Claims about "scaling laws" for many-body quantum states cannot be conclusively established without pushing toward larger system sizes where true asymptotic behavior emerges.

- The paper attributes the saturation of fidelity with increasing $N_{det}$ and $N_{layer}$ to limits in "representation power" (expressivity). However, this saturation could easily stem from optimization bottlenecks, such as vanishing gradients or poor parameter initialization in deeper networks. There is no analysis of gradient norms or optimization trajectories to isolate expressivity from trainability.

- While comparing FermiNet and PsiFormer is useful, the benchmark omits traditional Variational Monte Carlo baselines (e.g., standard Slater-Jastrow or simple pair-product states). Including these is necessary to contextualize whether deep neural networks actually offer a scaling advantage over classical trial wavefunctions on these specific targets.

---

> ### Author Rebuttal · Authors · 2026-03-31
>
> We are grateful for the reviewers on the validation of the solidity and presentation of this work. We have also included additional experimental data in the same anonymous GitHub link for reviewer’s verification. We have made targeted changes and new contents to reflect the weaknesses(W) and questions(Q) raised by the reviewer:
>
> W1. We have performed additional experiments on laughlin_m1 up to 28 electrons. We note our goal is to examine fidelity scaling within the representation capacity of the neural network. Current electron number is already sufficient to see the representation scaling and further increase in Ne will greatly increase the computational cost beyond our current GPU budget.
>
> W2. We thank the reviewer for this comment. To rule out the
> probabilities of local minimums, we have performed 5 independent optimizations
> with different random seeds on one of the hardest wave functions, moore_m6, and all training trajectories converged to the
> same value. This provides empirical evidence on the convergence of our results.
>
> W3. We thank the reviewer for this suggestion. We have tested the performance of a
> simple Slater-Jastrow ansatz on laughlin_m3 wavefunction. The simple ansatz has
> the following form:
> $\Psi = {\rm det} \left[ \sum_{n=1}^{N} C_{jn}\, z_i^{n}\, e^{-\frac{|r_i|^2}{4l_B^2}} \right] \exp\left[ \alpha \sum_{i<j} |\mathbf{r}_i - \mathbf{r}_j|^2 \right].$ At $N_e=2$, the Slater-Jastrow ansatz can capture partially the phase structure. However, at $N_e=3$, it fails to represent the nodal structure of the target laughlin_m3 wavefunction and can only reach a small value in fidelity.
>
> Q1. For size intensive ansatz (e.g. Full configuration interactions), the fidelity would decay exponentially and eventually vanish at large system size. However, it is possible to have a non-exponential scaling when the ansatz is size extensive, which is the case for NN wavefunctions (arxiv: 2602.02665, arxiv:2502.17144). In addition, for Pfaffian-based and superconducting wavefunctions, the minimal non-zero electron is 2. To ensure a consistent benchmark, we choose the empirical fitting function$ F = 1-\alpha(N_e-2)^\beta$, which can fit well for all wave functions included in our dataset.
>
> Q2. We have performed additional experiments to calculate the per-particle Coulomb energy of laughlin_m3 target wavefunction and psiformer&ferminet. We find that $E_{\rm coul}/ N_e$ matches closely for both NN wavefunctions. We choose fidelity as the benchmark metric, as it has known ground truth of 1 across all systems. This allows a direct comparison across different systems and sizes. This aligns well with our purpose, which is to provide a generic benchmark for NN wavefunctions. In fact, fidelity is commonly used as an expressivity benchmark in quantum state tomography (Phys. Rev. Research 6, 023250; Nature Physics 14, pp.447–450, 2018).
>
> Q3. To mitigate concerns on the trapping to local minima, we have performed 5 independent optimizations with different random seeds on one of the hardest wave functions, moore_m6. We find that all training
> trajectories converged to the same value. This provides empirical evidence on
> the numerical convergence of our results.
>
> Q4. In most general setting, the Monte Carlo variance would
> increases with the number of electrons. However, we make sure the network is initialized with relatively low variance. We also clip the gradient norm and damp the conditioners to restrict the impact of outliners during training. This allows us to maintain a stable $c^\ast$ across the
> training process.
>
> We thank the reviewer for the detailed review. We hope these clarifications adequately address the reviewer's concerns. We would truly appreciate it if the reviewer would consider raising the score. We are happy to answer any further questions during the discussion period.

---

> > ### Author Rebuttal · Reviewer_jUjv · 2026-04-02
> >
> > I thank the authors for their detailed rebuttal and the additional experiments provided. The theoretical clarifications regarding the scaling issue are also clear and satisfactory. All my questions have been properly resolved. Thanks for your hard work.

---

> > > ### Author Response · Authors · 2026-04-06
> > >
> > > We sincerely appreciate the reviewer’s positive assessment of our work and their thoughtful feedback. We also thank the reviewer for the time and effort spent in the review process and the follow-up discussions. This significantly helped us further improve the quality and clarity of the manuscript. We hope that our replies address the reviewer's concerns. and would truly appreciate it if the reviewer would consider increasing the overall evaluation of our work.

---

### Official Review · Reviewer_wgGJ · 2026-03-12

**Soundness:** 2
**Presentation:** 3
**Significance:** 4
**Originality:** 4
**Overall Recommendation:** 4
**Confidence:** 4

**Summary:**

The authors introduce a benchmarking dataset to benchmark the expressivity of neural quantum states for fermionic systems in the continuum. They include superconducting states, topological states and Wigner crystals. They benchmark the performance on Psiformer and Ferminet, two common architectures, tracking the fidelity as a function of electrons and number of determinants as well as layers. They derive scaling laws for the fidelity and find that Psiformer achieves better results than Ferminet across all target wave-functions.

**Compliance With Llm Reviewing Policy:**

Affirmed.

**Final Justification:**

The authors have addressed some of my concerns, but I am still not fully convinced by their results. I have updated my score accordingly.

**Key Questions For Authors:**

1) Why was an exponential decay envelope included for all topological wave-functions, and how are the resulting states still topological? I would expect the presence of topology to pose an obstruction to exponential localization?

2) The fidelity is not a very useful tool in studying the reliability of a variational wave-function with system size, due to the orthogonality catastrophy. Further, it does not give any indication of whether the obtained state is within the same quantum phase as the target state.
The relevant tools would be a set of well-defined observables of physical relevance that detect the crystalline/fluid nature of the state, its topological order and superconducting pairing mechanism. Have the authors tested this, and what are the scaling laws with respect to physical parameters?

3) Which boundary conditions are implemented for the benchmark wave-functions?

4) The training procedure is different than the standard task in VMC, minimization of a variational energy. How do you expect the difference in training landscape to affect the transferability of your results?

5) In comparing to basing the loss function on the fidelity for the benchmark, why did the authors choose to minimize $-\log(\mathrm{F})$ (with F the fidelity), instead of the more standard infidelity minimization?

6) Matching the current between wave-functions $\psi$ and $\phi$ is symmetric with respect of exchanging $\psi$ and $\phi$. However, Eq.~(12) breaks this symmetry by sampling over $|\Psi|^2$ only, which introduces a bias (for instance, zeros of  $|\Psi|$ won't be included while zeros of $\phi$ will). This bias is well-known in estimation e.g. the fidelity, where the usual approach to restore symmetry lies in estimating the fidelity by sampling from both wave-functions separately, and taking the square root of the product of the results.
How does the bias of non-symmetric sampling in the loss function affect the results?

**Limitations:**

yes

**Strengths And Weaknesses:**

Strengths:
The introduction of a benchmark dataset to benchmark the expressivity of neural quantum state architectures for condensed matter systems is highly relevant and much-needed. The dataset is extensive (see weaknesses for what is missing in my opinion) and available in the associated code package. The authors also present an advance in how to optimize towards a target wave-function, which has been challenging for continuum fermions and has practical implications e.g. for pre-training.

Weaknesses:
While the introduction of such a dataset is much-needed in the community, and I would like to see this paper published eventually in a good journal, in its current form there are some significant weaknesses. In particular, while the dataset is extensive, the associated code does not provide any interface to it - which, if it is the main take-away of this work, would be needed in my opinion. If the main take-away is not the dataset, but the actual performed benchmarks, such an interface may not be necessary, but at least code to reproduce the figures. In addition, the performed benchmark has many shortcomings, as I detail below.

Benchmark dataset:
The benchmark dataset overall is exhaustive and well-crafted. It is however missing liquid states, and intermediate correlated states. While a fully non-interacting Fermi liquid is trivial to represent, it would be still worth including it to make sure the network is able to pass that simple test. In addition, it would be good to include correlated states - i.e. a Fermi liquid with an optimal few-parameter Jastrow factor (of known form and parameters, see e.g. https://journals.aps.org/prb/pdf/10.1103/PhysRevB.16.3081), and a Wigner crystal with (2-body, thus transferable to larger system sizes) jastrow factor.

Soundness:
Benchmark procedure:
I am not convinced the fidelity is the correct quantity to consider to assess the approximation abilities of the neural network: In particular since the goal is to observe its scaling with system size, the orthogonality catastrophy pretty much limits the usefulness of the fidelity to indicate accuracy as a function of system size. Even at same system size, it is fine to consider the fidelity as one measure, but the actually relevant question is whether the neural-network wave-function has the same (topological etc.) order parameters as the state it tries to represent, and to which precision physical observables are matched. For a useful benchmark dataset, I would thus expect including a collection of such observables.

In addition, the benchmarks maximally reach 19 electrons, which is comparably very small to make any statements about the physical nature of the considered states.

Code:
The associated code package does not allow reproducing any figure or results. It just contains the coded wave-function, but no part of the benchmark and optimization procedure.

Presentation:
The section "Related work" is somewhat redundant and not quite to the point. First, the paragraph "NN wavefunction expressivity" is doubled: The untitled paragraph below contains exactly the same content, just slightly rephrased. Further, there are some explanations of work that seems unrelated. For instance, the sentence "In (Nazaryan et al., 2025), neural-network wavefunctions are first pretrained to match a chosen trial state. Starting from this initialization, the authors further optimize the network with respect to the fractional quantum Hall Hamiltonian, and demonstrate accurate ground state results for systems with Coulomb interactions and Landau-level mixing" does not seem to address a "systematic expressivity scaling of NN wavefunctions" and could use some clarification.

Smaller comments:
- The naming of the hyperparameter $\alpha$ in Eq.~(10) is confusing, as $\alpha(R)$ is already defined. It is then re-introduced a third time on page 6, by approximation of the power law scaling of the fidelity

---

> ### Author Rebuttal · Authors · 2026-03-31
>
> We are grateful for the reviewers on the validation of the content of this work, and are thankful to the constructive comments. We have also included additional experimental data in the same anonymous GitHub link for reviewer’s verification. We have made targeted changes and new contents to reflect the weaknesses(W) and questions(Q) raised by the reviewer:
>
> W1. Benchmark dataset:
>
> We agree that including such wavefunctions could further expand the diversity of our dataset. We have tested the fidelity scaling of a target Fermi Liquid wavefunction with sine&cosine orbital and Mc Millan type correlator (PhysRevB.16.3081). We find psiformer can reach F=0.94 at Ne=13. It indicates the difficulty of Fermi Liquid for psiformer is between topological wavefunctions and Wigner crystal.
>
> W2. Benchmark procedure:
>
> For size intensive ansatz (e.g. Full Configuration Interaction), the fidelity would vanish when the system size grows. However, it is possible to have non-vanishing overlap when the ansatz is size extensive, which is the case for NN wavefunctions (arxiv: 2602.02665, arxiv:2502.17144).
>
> We choose fidelity as the metric, as it has known ground truth of 1 across all systems. This allows a direct comparison across different systems and sizes. This aligns well with our purpose, which is to provide a generic benchmark for NN wavefunctions. In fact, fidelity is commonly used as an expressivity benchmark in quantum state tomography (Phys. Rev. Research 6, 023250; Nature Physics 14, pp.447–450, 2018).
>
>
> We have added calculations for various observables.  For topological wavefunctions, we calculated the phase structure, which is related to the topological observables. For superconducting wavefunctions, we show the pair density matrix, which reveals superconducting condensation. For Wigner crystals, we show the charge density, which directly visualizes the symmetry breaking patterns. We find that higher fidelity correlates positively with observable accuracy. This can be seen by: $| \langle \psi | O | \psi \rangle - \langle \phi | O | \phi \rangle | \le 2 ||O|| \sqrt{1 - F}$ for any bounded operator $O$.
>
> W3. Electron number: We have performed additional experiments on laughlin_m1 up to 28 electrons. We note our goal is to examine fidelity scaling within the representation capacity of the neural network. Current electron number is already sufficient to see the representation scaling and further increase in Ne will greatly increase the computational cost beyond our current GPU budget.
>
> W4. Code availability: We have added the loss functions into the anonymous GitHub link.
>
> W5. Presentation: We have revised the related phrasings and improved comprehensibility of the related work section.
>
> Q1.  The topological wavefunctions used are defined on a disk from lowest Landau-Level (LLL) projections. The exponential envelope originates from the projection to LLL and reflects the physical prior of the system. This is a standard practice and has been used in previous works (PhysRevB.111.205117).
>
> Q2. Please refer to our response to W2 on reviewer’s the question on fidelity and observables.
>
> Q3. The boundary conditions we used follow standard practice in the field. The topological wavefunctions are defined on a disk. The superconducting and Wigner crystal wavefunctions are defined on a torus with periodic boundary conditions.
>
> Q4. Indeed, differences in the training landscape exist in the two procedures. However, we ensure the training is converged in this work. We performed additional training with different random seeds, and all training trajectories converged to the same value. We expect our results on network expressivity should also be transferable to VMC training, as high fidelity indicates a bounded energy difference (for the same reason as our response on operator accuracies). We also include an additional experiment that the coulomb energy between the target laughlin_m3 wavefunction and psiformer matches closely.
>
> Q5. -logF is numerically more stable than F and is a standard training practice in the field(NeurIP 36 pp. 450-476, 2023; Phys. Rev. R, 5, 013216). We also note that -log(F) would behave the same as 1-F when F is close to 1.
>
> Q6. The bias has a small effect on the performance of Eq.~12. During the pretrain stage, $L_{\rm prob}$ is activated before $L_{\rm cur}$. This is achieved by tuning hyperparameter alpha. We choose an alpha that varies sufficiently slowly such that, when $L_{\rm cur}$ is activated, $L_{\rm prob}$ is sufficiently optimized and $|\psi|^2$ is very similar to $|\phi|^2$.
>
>
> We thank the reviewer for the detailed review. We hope these clarifications adequately address the reviewer's concerns. We would truly appreciate it if the reviewer would consider raising the score. We are happy to answer any further questions during the discussion period.

---

> > ### Author Rebuttal · Reviewer_wgGJ · 2026-04-03
> >
> > The authors have answered some of my questions. However, I am not convinced the answers are yet sufficient to raise the score. A few comments and questions:
> >
> > - I’m still not convinced the scaling of the fidelity is a meaningful quantity. Quantum state tomography is usually performed on much smaller system sizes, and not in the continuum. The bound for the observables, however, is more convincing – the computed observables in the paper currently don’t show any scaling law, but if there was a section using the bound in a meaningful way to derive a scaling law, it would be more convincing
> > - The boundary conditions on a disk for topological wave-functions are not standard practice, it has just been done in a few works because it is easier. There is no physical reason for it. Clearly stating this in the paper would be necessary. Same with the gaussian envelope, to make clear it is the gaussian on the disk and not around eventual wigner crystal centers
> >
> > I don’t find the references NeurIP 36 pp. 450-476, 2023; Phys. Rev. R, 5, 01321
> >
> > Can you give links? Also, it is not standard practice in the field. Infidelity minimization has a long history in VMC-related context.

---

> > > ### Author Response · Authors · 2026-04-06
> > >
> > > We thank the reviewer for the follow-up questions.
> > >
> > > **fidelity as a meaningful quantity**
> > >
> > > We understand the reviewer's concern. We agree that the correlation between observables and fidelity is important to the implication of our results in different scenarios. To address such concern, we performed an additional experiment to show that, as fidelity grows during the training, the expectation values of kinetic energy and coulomb energy gradually match between the target and NN wavefunctions (the calculation below is performed on wc_gau_v1A at Ne = 9):
> > > Step  | Fidelity  | $\Delta E_{kin}$ (%)  | $\Delta E_{coul}$ (%) | $\Delta E_{total}$ (%)
> > > ------|-----------|------------------------------|---------------------------------|-------------------------------
> > > 0     | 0.0009  | 31.62                      | 10.26                         | 20.36
> > > 20     |  0.1844  | 29.35                      | 4.01                          | 20.32
> > > 60      | 0.7006  | 4.95                       | 0.75                          | 3.83
> > > 200      | 0.9440  | 3.97                       | 0.08                          | 2.93
> > > 5000     | 0.9981  | 1.59                       | 0.07                          | 1.14
> > >
> > > This indicates a positive correlation between infidelity and observable/energy error. A complete plot can be found in the same github link. We also note that quantum state tomography has been recently extended to large system with hundred qubits with fidelity as a metric (Nature Physics 16, 1050-1057) and gradually developed for continuous variable system (Nature Physics 21, 2002–2008; PRX Quantum 5, 010346). While we agree that fidelity does not provide all the information of a system, it serves as a useful metric in quantum many-body physics and information theory.
> > >
> > > While there is correlation and bound between fidelity and observable error,  we do not claim that a universal observable scaling law can be extracted from such correlation. We would like to point out an important difference between the purpose of this work and the study of observable scaling law. Our goal is to provide a benchmarking protocol that allows direct comparison across systems with different physical origins. For different systems, the physically relevant observables also differ. It is proven in quantum information and classical shadows theory that different observables require different measurement and algorithms for accurate learning. Therefore, a comprehensive study of precise observable scaling laws would require observable-specific loss functions and carefully chosen physically relevant observables, inevitably leading to different training algorithms and reduced transferability across different systems. It also remains unclear whether a common scaling law exists for different observables, as the error behavior can vary significantly between different types of observables. For example, the error in the expectation value of a one-body operator may differ from that of more non-local quantities.
> > >
> > >
> > > We agree with the reviewer on the importance of observable scaling laws. While a systematic investigation is beyond the scope of the current work, we view this as an important direction for future research. Meanwhile, we will add discussion on the relationship between observable errors and fidelity to address the reviewer’s concern on its validity as a scaling metric.
> > >
> > > **disk boundary condition**
> > >
> > > We agree with the reviewer that different boundary conditions can be used to study topological wavefunctions. The disk geometry considered in the referenced PRB work by other authors represents only one of several possible choices. We will clarify this point and include additional discussion in the revised manuscript.
> > >
> > > We will also revise the manuscript to clarify that the Gaussian factor arises from the lowest Landau level structure and make clear that it is the gaussian on the disk and not around eventual Wigner crystal centers.
> > >
> > > **-log(F) as loss function**
> > >
> > > We apologize for the unclear reference in our previous reply. The arXiv version of the two references can be found at arXiv: 2304.01996; arXiv: 2101.07243.
> > >
> > > We agree that infidelity optimization is commonly used. In our context, we note that -log F and infidelity have similar performance, as $\partial_\theta {\rm log}(F) = \frac{1}{F} \partial_\theta F$. Therefore, the performance of -log(F) and 1-F are similar when F is close to 1, and the $1/F$ factor can be absorbed into the learning rate. We also provide additional references that are related to using -log(F) as loss functions: arxiv:1808.05232; Comput. Phys. Commun. 300, 109169.
> > >
> > > We appreciate the reviewer’s additional comments, which has significantly helped us to improve the quality of our work. Thank you for your time in reviewing our work. We hope that the above replies address the reviewer's concerns. and would truly appreciate it if the reviewer would consider increasing the overall evaluation of our work.

---

### Official Review · Reviewer_FUia · 2026-03-13

**Soundness:** 3
**Presentation:** 3
**Significance:** 3
**Originality:** 2
**Overall Recommendation:** 5
**Confidence:** 3

**Summary:**

This paper introduces a new dataset and evaluation pipeline for neural-network wavefunctions for strongly correlated systems, including topological states, Wigner crystals, and superconducting wavefunctions. The authors propose to use wavefunction fidelity as the evaluation metric and benchmarked the Ferminet and Psiformer models.

**Compliance With Llm Reviewing Policy:**

Affirmed.

**Final Justification:**

My concerns are fully addressed during rebuttal.

**Key Questions For Authors:**

- Why use the fidelity instead of the VMC energy as the evaluation metric?
- In Figure 3, why does Ferminet fail for Moore_m6?
- What are physics contexts or applications for the selected systems?
- How do the number of evaluation burn-in steps affect the results?

**Limitations:**

Yes.

**Strengths And Weaknesses:**

Strength

- The paper is clearly presented.
- The motivation to have a benchmark and unified evaluation is well-founded.

Weakness

- The number of electrons is not very large (<20), it is not very clear why scaling law matters in this regime.
- The results for Wigner crystals seem almost at 100% (e.g., in Figure 3). Does this mean the benchmarked systems can already be solved by existing methods?
- The tested systems have known ground truth, which is good for benchmarking purposes, but does this also mean the tested systems are small enough to be solved exactly?

---

> ### Author Rebuttal · Authors · 2026-03-31
>
> We are grateful for the reviewer's encouraging comments on the presentation and motivation of our work. We have also included additional experimental data in the same anonymous GitHub link for reviewer’s verification. Targeted changes and new contents are made to reflect the weaknesses(W) and questions(Q) raised by the reviewer.
>
> W1. We thank the reviewer’s comments. We have performed additional experiments on laughlin_m1 up to 28 electrons. We note our goal is to examine fidelity scaling within the representation capacity of the neural network. Current electron number is already sufficient to see the representation scaling and further increase in Ne will greatly increase the computational cost beyond our current GPU budget.
>
> W2&W3. We thank the reviewer for the question. One of the goals of our work is to test the representational hardness for different wavefunctions. Our results indeed show that the Wigner crystal wavefunction is an easier representational task compared with other wavefunctions. But it doesn’t show that the system can be solved exactly. As the system size grows, the fidelity still demonstrates a non-trivial scaling with respect to electron number for Wigner crystal wavefunctions.
>
> Q1. We thank the reviewer for the comment. The purpose of this work is to provide a generic representational benchmark for neural network wavefunctions. Therefore, we choose fidelity as the evaluation metric, as it has a known ground truth of 1 and is invariant to system sizes. This allows a direct comparison across different systems and sizes. On the other hand, VMC energies have different scales across different systems, and the ground truth is usually inaccessible.
>
> Q2. Moore_m6 is a highly frustrated wavefunction with complex phase structures and windings due to the large value of m. Ferminet also has weaker expressive power compared with psiformer. During the pretrain stage, Ferminet’s expressivity is insufficient to capture enough phase structure, and the following fidelity training has a trivial signal and fails converging to a meaningful value.
>
> Q3. We thank the reviewer for this question. We include target wavefunctions from three major classes—topological states, superconducting states, and Wigner crystals—covering a broad range of physically important systems with intrinsically different correlation structures. The topological class includes well-known trial wavefunctions that originated from the fractional quantum Hall systems projected to lowest Landau-Levels. They are found in many topological materials such as rhombohedral graphene and transition metal dichalcogenides. The superconducting class includes BCS–type wavefunctions, which are found in many superconducting materials. Two famous examples would be cuprate and nickelate. The Wigner-crystal class includes states that exhibit spontaneous symmetry breaking in contrast to conventional Fermi gas, which has been seen in quantum materials where interaction energy dominates kinetic energy.
>
> Q4. We thank the reviewer for this question. The number of evaluation burn-in steps have a limited effect on the results. This is because the Markovian chain is fully thermalized during the training stage. We directly use the fully thermalized walkers to evaluate the final fidelity.
>
> We thank the reviewer for the detailed review. We hope these clarifications adequately address the reviewer's concerns. We would truly appreciate it if the reviewer would consider raising the score. We are happy to answer any further questions during the discussion period.

---

> > ### Author Rebuttal · Reviewer_FUia · 2026-04-04
> >
> > Thank you for the detailed responses. The rebuttal clarified my questions about scaling law, physical context and experimental results (although it still seems strange that only one system failed. I have some follow-up questions below:
> >
> > - For Moore_m6 using FermiNet, have you tried training multiple times (so they can have different initializations)? Would some tweaks be able to make the training work?
> >
> > - What are the computational costs for the experiments (e.g., GPU memory usage and training time)?
> >
> > - It is true that fidelity has a ground truth of 1 for all systems, so it is provides a unified metric. Is this advantage strong enough to support it as the main metric? For example, FermiNet and Psiformer papers mainly benchmarked VMC energy for molecules. Can a good fidelity reflect a good VMC energy as well?

---

> > > ### Author Response · Authors · 2026-04-06
> > >
> > > We thank the reviewer for the follow-up questions.
> > >
> > > **Moore_m6 training using FermiNet**
> > >
> > > We tried to train the network with different initializations (random seeds). However, during the pretrain stage, $L_{\rm cur}$ fails to converge to a sufficiently low value regardless of the choice of random seed, which leads to a vanishing signal during the following consecutive fidelity training. We rerun the same training (moore_m6 at $N_e = 8$ with ferminet) using different random seeds and with longer training steps (30k steps). The value of $L_{\rm cur}$ at 20k steps (default setting used in the paper) and at 30k steps can be found below:
> > > Model |$L_{\rm cur}$(at 20k steps) ↓  | $L_{\rm cur}$(at 30k steps) ↓
> > > ----|----------------------|----------------------
> > > Ferminet(seed 0)   | 6.14 |  5.66
> > > Ferminet(seed 1) | 6.24  | ​​ 6.14
> > > Ferminet(seed 2) | 6.89  |  6.30
> > > Psiformer | 1.58 | 1.48
> > >
> > > **Computational Cost**
> > > The memory scaling depends on architecture and model size. For all system sizes and network sizes calculated in this work (including the Ne =28 case shown in the additional experiment), our calculations can fit into a single 80Gb H100 GPU. We provide the following table for training time and memory spent on the bcs_s wavefunction. The training is performed on 1 H100 GPU. The time (memory) reported is in the unit of seconds/100 steps (MiB).
> > >
> > > Ne | PsiFormer (time / memory) | FermiNet (time / memory)
> > > ---|---------------------------|--------------------------
> > > 2  | 5.12  (1640)              | 6.15  (2026)
> > > 4  | 9.58  (2280)              | 11.89 (2540)
> > > 6  | 16.16 (3306)              | 25.34 (4076)
> > > 8  | 25.53 (2534)              | 35.97 (4076)
> > > 10 | 39.65 (3558)              | 48.52 (4076)
> > > 12 | 55.57 (3560)              | 70.86 (6140)
> > > 14 | 78.83 (2818)              | 105.11 (6140)
> > > 16 | 96.87 (4074)              | 144.60 (7614)
> > >
> > > We note that memory usage does not scale monotonically with Ne due to the internal kernel fusion by Jax XLA, and we leave a systematic study of memory scaling to future work.
> > >
> > > **fidelity as unified metric**
> > > Fidelity is strictly bounded between 0 and 1. The ground truth, that is the optimal achievable fidelity is therefore 1 regardless of the system type and sizes. We note that quantum state tomography has been recently extended to large system with hundred qubits with fidelity as a metric (Nature Physics 16, 1050-1057) and gradually developed for continuous variable system (Nature Physics 21, 2002–2008; PRX Quantum 5, 010346). Even though VMC energy is an important quantity, it may pose challenges for universal benchmarking because its scale highly depends on the type of physical system. For example, the GS energy of a spin liquid would differ to that of a Chern insulator. Although empirical quantities exist, they are largely affected by the low lying energy gaps of the system and may vary across systems
> > >
> > > In addition, solving the exact ground truth to VMC energy is known to be QMA-hard and cannot generally be obtained in polynomial time due to the exponentially large Hilbert space. This makes exact ground truth benchmarks impossible above a certain system size threshold.
> > >
> > > **Can a good fidelity reflect a good VMC energy as well?**
> > >
> > > We thank the reviewer for this question. We note that infidelity correlates positively with energy error. As evidence, we show in the table below that, as fidelity becomes larger during the training, the expectation values of kinetic energy and coulomb energy gradually match between the target and NN wavefunctions. The calculation below is performed on wc_gau_v1A at $N_e = 9$:
> > > Step  | Fidelity  | $\Delta E_{kin}$ (%)  | $\Delta E_{coul}$ (%) | $\Delta E_{total}$ (%)
> > > ------|-----------|------------------------------|---------------------------------|-------------------------------
> > > 0     | 0.000965  | 31.6277                      | 10.2663                         | 20.3654
> > > 20     |  0.184419  | 29.3531                      | 4.0150                          | 20.3254
> > > 60      | 0.700679  | 4.9596                       | 0.7521                          | 3.8319
> > > 200      | 0.944042  | 3.9784                       | 0.0873                          | 2.9305
> > > 5000     | 0.998174  | 1.5901                       | 0.0711                          | 1.1426
> > >
> > > A complete plot can be found in the anonymous github link provided in the manuscript. We also note that the observable error is bounded by fidelity: $| \langle \psi | O | \psi \rangle - \langle \phi | O | \phi \rangle | \le 2 ||O|| \sqrt{1 - F}$ for any bounded operator O. We will add related discussion between observable error and fidelity in the revised manuscript
> > >
> > > We appreciate the reviewer’s additional comments, which has significantly helped us to improve the quality of our work. Thank you for your time in reviewing our work. We hope that the above replies address the reviewer's concerns. and would truly appreciate it if the reviewer would consider increasing the overall evaluation of our work.

---

### Decision · Program_Chairs · 2026-04-30

**Decision:**

Accept (regular)

**Comment:**

The authors present a series of benchmarks for neural network wavefunctions for correlated electronic systems in real space. The reviewers agreed this was a useful and valuable contribution to the field of neural network wavefunctions. It is in many ways the real-space analog of what Wu, Rossi et al. did in their 2024 Science paper for lattice models. While some authors had concerns that the benchmarks were for relatively small systems (up to 28 electrons) the consensus was that this was an important contribution to the field. I recommend acceptance.